# Classification of Diabetic Walking for Senior Citizens and Personal Home Training System Using Single RGB Camera through Machine Learning

Yeoungju Woo [1,†], Seoyeong Ko [1,†], Sohyun Ahn [1,†], Hang Thi Phuong Nguyen [1,†], Choonsung Shin [2], Hieyong Jeong [1,*], Byungjoo Noh [3], Myeounggon Lee [4], Hwayoung Park [5] and Changhong Youm [5,*]

1   Department of Artificial Intelligence Convergence, Chonnam National University, 77 Yongbong-ro, Bukgu, Gwangju 61186, Korea; 218860@jnu.ac.kr (Y.W.); 175847@jnu.ac.kr (S.K.); 170186@jnu.ac.kr (S.A.); 208258@jnu.ac.kr (H.T.P.N.)
2   Graduate School of Culture, Chonnam National University, 77 Yongbong-ro, Bukgu, Gwangju 61186, Korea; cshin@jnu.ac.kr
3   Department of Kinesiology, Jeju National University, Jeju-si 63243, Korea; bnoh@jejunu.ac.kr
4   Center for Neuromotor and Biomechanics Research, Department of Health and Human Performance, University of Houston, Houston, TX 77004, USA; mlee47@central.uh.edu
5   Department of Health Sciences, The Graduate School of Dong-A University, Saha-gu, Busan 49315, Korea; app00113@dau.ac.kr
*   Correspondence: h.jeong@jnu.ac.kr (H.J.); chyoum@dau.ac.kr (C.Y.); Tel.: +82-625-303-427 (H.J.); +82-512-007-830 (C.Y.)
†   These authors contributed equality to this study.

**Abstract:** Senior citizens have increased plasma glucose and a higher risk of diabetes-related complications than young people. However, it is difficult to diagnose and manage elderly diabetics because there is no clear symptom according to current diagnostic criteria. They also dislike the invasive blood sample test. This study aimed to classify a difference in gait and physical fitness characteristics between senior citizens with and without diabetes for a non-invasive method and propose a machine-learning-based personal home-training system for training abnormal gait motions by oneself. We used a dataset for classification with 200 over 65-year-old elders who walked a flat and straight 15 m route in 3 different walking speed conditions using an inertial measurement unit and physical fitness test. Then, questionnaires for participants were included to identify life patterns. Through results, it was found that there were abnormalities in gait and physical fitness characteristics related to balance ability and walking speed. Using a single RGB camera, the developed training system for improving abnormalities enabled us to correct the exercise posture and speed in real-time. It was discussed that there are risks and errors in the training system based on human pose estimation for future works.

**Keywords:** automated machine learning; diabetic walking; elderly diabetics; human pose estimation; machine-learning-based personal home training system

## 1. Introduction

Diabetes is a metabolic disease that causes problems with either the secretion or assimilation of insulin, a natural hormone in which the function is to reduce the sugar concentration in the blood [1]. If diabetes is not managed, high blood sugar levels and other risk factors can lead to the blood vessel and nerve damage [2]. Moreover, the complications of diabetes can develop and affect nearly every organ system in the body. In particular, we have large and small blood vessels that deliver blood around the body. Damage to the large blood vessels leads to heart attacks, several kinds of strokes, or affects blood flow to the lower extremities, and risk to the small blood vessels can affect the eyes, kidneys, teeth and gums, and nerves. In addition, nerve damage can affect the digestive system,

sexual organs, and excretory system. That is the significant problem as to why there is no complete cure for diabetes yet. Current treatments for diabetes are only to check the amount of glucose in the blood, adjust the food, and keep exercise activities every day by oneself [1,3].

Walking is the activity recommended for most diabetic patients, while being effective in weight loss and maintenance and in improving glucose control [4]. This recommendation comes from the results of a meta-analysis, including many small, short-term randomized controlled clinical trials (RCTs) and some recent additional research, showing clinically appreciable improvement of HbA1c [5,6]. The effect on insulin resistance is not apparent. Walking is easily applicable in daily life in most patients without requiring expertise, and logistic support can be performed in different places [7,8]. Limited information indicates the improvement of several alterations involved in the increased cardiovascular risk associated with diabetes. Moreover, a small study reported favorable changes in several functional aspects of diabetic neuropathy, although the presence of this condition requires specific monitoring of patients and may also limit walking activities [4].

Many researchers have shown that elders with diabetes were related to a greater risk of falls, and this was more clear in insulin-treated patients [9]. For example, according to six studies involving 14,685 participants, the number of falls in diabetic and non-diabetic, respectively, was 25.0%, and 18.2% [10]. On the other hand, diabetes increased 94%, and 27% risk of falls in insulin-treated and no-insulin-treated patients, respectively [11]. Hence, preliminary screening before starting any physical activity programs in older adults with diabetes mellitus should include a general medical examination, with specific attention to symptoms and signs of chronic complications (cardiovascular disease, nephropathy, retinopathy, and neuropathy), and assessment of metabolic control [12]. However, the significant problem is that there are large changes in the analysis results due to the diverseness of the diabetic population with the presence and severity of diabetes complications. Accordingly, although the prescription of walking in patients with diabetes should be preceded by a tailored medical and functional assessment, it is difficult to assess the functionality of walking capacity by themselves [13]. Thus, senior citizens with diabetes must recognize the diabetic walking abnormality for self-preservation and evaluate the functionality without expert medical knowledge using the simple sensing system.

The results of several studies have been conducted to assess walking biomechanics alterations in diabetic neuropathic subjects [11]. According to kinetics and muscle activation patterns, there have been significant variations in the results of both a reduction and an increase in the gastrocnemius activity and the lower limb joints moments. In terms of plantar pressure [14,15], a shorter center of pressure (CoP) excursion and a higher peak pressure over the forefoot have been found [16]. Although the diabetic symptoms are apparent, the neuromuscular, kinematics, and kinetics changes do not show a distinct pattern associated with the results of diabetes and diabetic neuropathy (DPN) [17–23]. Compared to these analyses of movement symmetry, continuous relative phase (CRP) is one of the most sensitive analyses for detecting the mutual relationship among the joints and asymmetries in coordination during walking, particularly for identifying cyclic movement deviations caused by diabetes [24,25]. However, because CRP also tries to explain biomechanical walking patterns, such as ground reaction forces, angles, and moments of the trunk, hip, knee, and ankle, the assessment should be performed at the well-equipped hospital with expert medical knowledge by using the expensive motion capture system. Thus, it seems complicated to protect diabetic senior citizens by themselves in daily life through conventional works.

Therefore, this study aimed to classify a difference in gait and physical fitness characteristics between senior citizens with and without diabetes through automated machine learning (AutoML) by using the simple sensor and propose a machine-learning-based personal home training system for training abnormal gait motions by oneself. For this study, a dataset was constructed by 200 senior citizens over 65 years old who performed to walk a flat and straight pathway of 15 m under three different conditions of walking speed

(slow (=20% slower than preferred walking speed), preferred, and fast (=20% faster than preferred walking speed)) by using an inertial measurement unit (IMU) and physical fitness tests. Then, for training abnormal gait motions, the proposed home training system enabled us to correct the exercise posture and speed in real-time through machine-learning-based similarity evaluation between training experts and novices by using a single RGB camera.

The contribution of this study was to make clear the association between diabetic walking abnormalities and measured feature vectors. Then, these results could be applied for early detection and therapeutic intercessions that rehabilitate the walking function in diabetic senior citizens by grouping them not only by clinical features but also based on their motor control strategies. Furthermore, the proposed home training system helps senior citizens with diabetes continue to exercise activities with the correct posture and speed.

The structure of this study is: we describe how to construct the dataset for this study in Section 2, then explain how to classify through AutoML in Section 3, and show all of the results, including which gait characteristics are essential to classify, in Section 4. Finally, to train the detected abnormality, we introduce the developed machine-learning-based personal home training system in Section 5 and show all of the results to verify the usefulness in Section 6. Finally, we discuss the classification for healthcare through AutoML and the limitations of the developed home training system in Section 7, before our conclusions in Section 8.

## 2. Dataset Construction

### 2.1. Human Subjects

Human subjects for this study participated in community activities in the Busan metropolitan city from 2018 to 2019. A total of 200 were recruited for human subjects aged 65 years old and over living in the community; 59 were in the group of healthy subjects, and the remaining 141 were in the group of subjects with diabetes. In the group for subjects with diabetes, there was no one to take anti-diabetes drugs to control the blood sugar level because their symptoms were not severe. Thus, the treatment of diabetes in this study was centered on diet, exercise, and weight loss. In addition, a human subject was excluded if he or she could not walk without any aid tool, had a history of severe orthopedic problems, or had neurosurgical and neurophysiological problems in the preceding six months.

Figure 1 summarizes the characteristics of two different groups for this study. There is no significant difference in age and body mass index (BMI) between subjects with diabetes and without diabetes. Although the number of females is more significant than that of males, the ratio of gender between two different groups is similar. There is no problem because the mean age for human subjects is 74 years old. The ratio of 80~91% for human subjects has an experience of compulsory education courses, including the elementary, middle, and high school.

| | Subjects without diabetes | Subjects with diabetes |
|---|---|---|
| Age [years old] | $74.7 \pm 5.7$ | $74.4 \pm 5.2$ |
| BMI [$^{kg}/_{m^2}$] | $24.6 \pm 3.1$ | $24.3 \pm 2.3$ |
| Gender [persons] | | |
|     Male | 22 (37%) | 52 (37%) |
|     Female | 37 (63%) | 89 (63%) |
| Education level [persons] | | |
|     Elementary | 17 (30%) | 53 (39%) |
|     Middle | 15 (26%) | 37 (26%) |
|     High | 14 (24%) | 37 (26%) |
|     University | 3 (5%) | 8 (6%) |
|     etc | 9 (15%) | 4 (3%) |

**Figure 1.** Characteristics of two different groups for this study.

If all participants agreed to attend all experiments, they had to read and sign the informed consent document approved by the Institutional Review Board of Dong-A University (IRB number: 2-104709-AB-N-01-201808-HR-023-02). All experimental procedures were performed under the Declaration of Helsinki.

### 2.2. Experiment for Data Acquisition

Figure 2 shows the experimental environment for the measurement and analysis phases under steady-state conditions. (a) Three arrows, from left to right side, indicate acceleration, consecutive, and deceleration steps for the measurement phase, respectively. (b) Detection of walking abnormalities with the shoe-type inertial measurement unit (IMU) system is analyzed through extracted features, such as heel strike (HS) and toe-off (TO) [26,27].

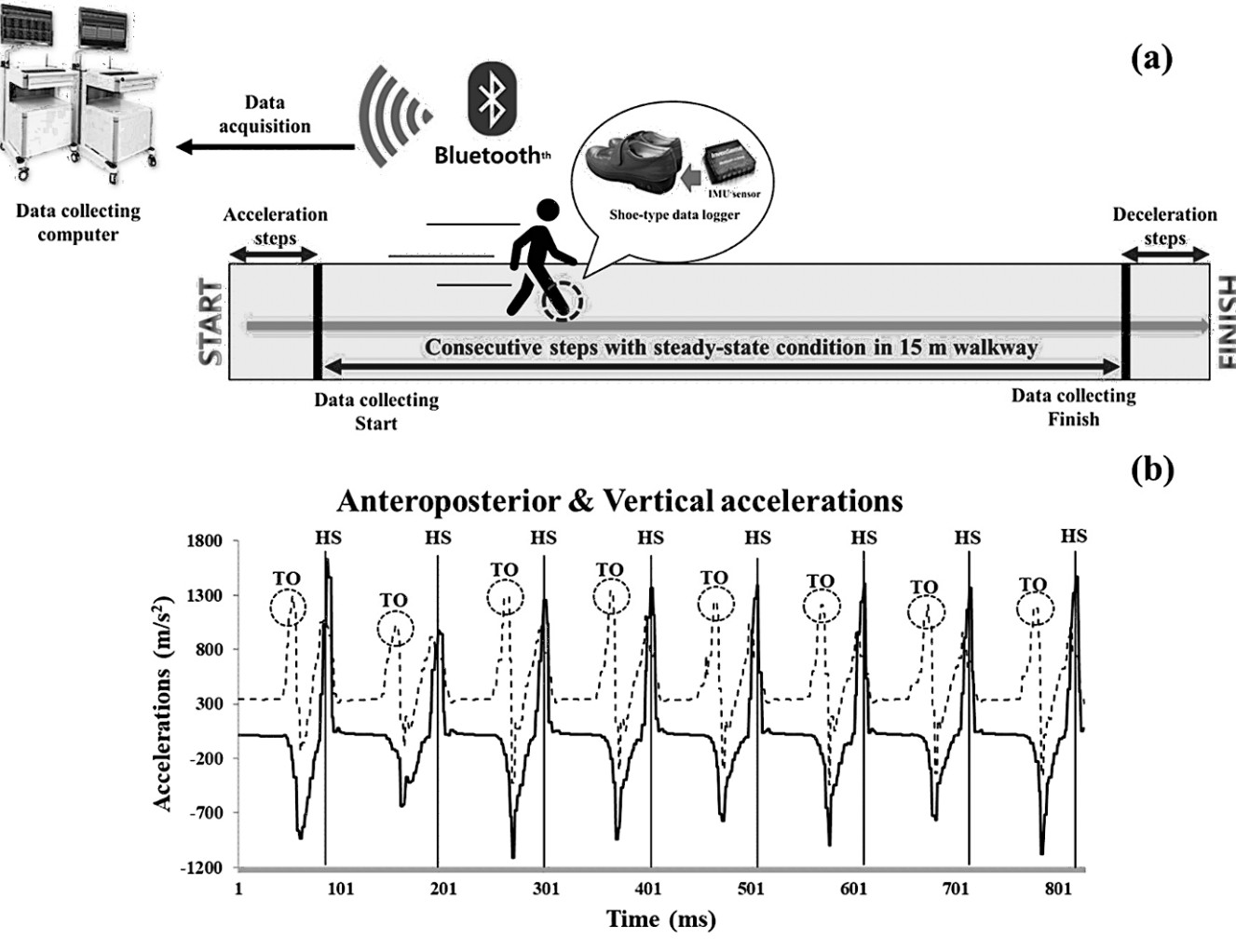

**Figure 2.** Experimental environment and system for the measurement and analysis phases under steady-state conditions. (**a**) Three arrows from left to right sides indicates acceleration, consecutive, and deceleration steps for the measurement phase, respectively. (**b**) Detection of walking abnormalities with the shoe-type inertial measurement unit (IMU) system is analyzed through extracted features.

Shoe-type IMU sensor (DynaStab$^{TM}$, JEIOS, Busan, Korea) consists of the data logger (Smart Balance SB-1, JEIOS, Busan, Korea) and the data acquisition device. The IMU sensor (IMU-300$^{TM}$, InvenSense, San Jose, CA, USA) in the data logger can measure triaxial acceleration (up to $\pm 6$ g) and tri-axial angular velocities (up to $\pm 500^\circ$ s$^{-1}$) along the three

orthogonal axes. The IMU sensors are set up on the outsoles of both shoes, and data are transmitted to the data acquisition device via Bluetooth. The walking measurement was collected at 100 Hz and filtered using a second-order Butterworth low-pass filter with the 10 Hz cut-off frequency. Although data are measured during acceleration and deceleration steps, data for these steps are ignored. Instead, data during consecutive steps are used to analyze for extraction of features.

All subjects performed three trials of the overground walking test along the straight 15-m walkway at slower, preferred, and faster speeds wearing the shoe-type IMU sensor. The preferred walking speed is defined by someone's comfortable, stable, and usual walking speed. The subjective decision of subjects decides the slower and faster speed. Thus, the slower or faster speed is controlled 20% slower or faster under the preferred speed. The prepared metronome supports the decision of the walking speed. The subjects were asked to walk at the preferred speed to measure cadence using the metronome before each test.

All subjects also performed four physical fitness domains with nine tests: muscle strength, flexibility, balance, and cardiorespiratory fitness. All subjects were performed grip strength with a hand-grip dynamometer (TKK 5401 Grip-D, Takei Scientific Instruments, Tokyo, Japan) and biceps curls with a dumbbell (3 kg for men; 2 kg for women) to measure upper extremity strength. Five times sit-to-stand (STS) and standing time from a prolonged sitting position were also performed to assess the lower extremity strength. To assess flexibility, back scratch as the upper extremity flexibility and chair sit and reach as the lower extremity flexibility was performed. Single-leg balance (dominant leg) as the static balance and a three-meter timed-up-and-go (TUG) as the dynamic balance were performed to assess physical abilities. Finally, a 6-min walk test was performed to assess cardiorespiratory (or functional) fitness. Two attempts of each test calculated the mean scores of physical fitness tests.

*2.3. Dataset*

The dataset consists of features extracted by the non-invasive method, such as the survey and measurement with the shoe-type IMU sensor. The average age of human subjects in this study is 74 years old, thus being senior citizens. It is thought that the experiment through the invasive method is complex because participants' burden is so enormous. So, it is crucial to extract the feature point for classification through the non-invasive method. The total number of features is 43 for the training and prediction, 7 for the survey, and 36 for the measurement.

At first, all human subjects were surveyed to check their health condition. Figure 3 shows the walking-related features extracted by the non-invasive method, including the survey and measurement. Extracted features with the survey are shown at the numbers 1, 2, and 3; Age, Gender, Education level, BMI, MET-min/week, Hypertension, MMSE score, Insomnia score, and Quality of Life (QOL). It is well-known that these features are famous for estimating personal life patterns, in general. The total number of extracted features was 9.

Then, all human subjects attended the walking experiment by wearing the shoe-type IMU sensor. Extracted features with the measurement are from the numbers 4 and 5, as shown in Figure 3; walking speed, stride length, CV stride length, CV stance phase, gait asymmetry, cadence, stride time, CV stride time, grip strength, five times sit-to-stand (STS), biceps curl, chair sit and reach, three-meter TUG, back scratch, single leg balance, six min walk, and standing time from a long sitting position. It is well-known that these features are also famous for evaluating walking and physical fitness characteristics, in general. All walking experiments occurred under the three different conditions of slower, preferred, and faster speeds. The total number of extracted features was 36.

| No | Parameters | Features | Description |
|---|---|---|---|
| 1 | Covariate | Age | |
| | | Gender | |
| | | Education level | |
| | | BMI | |
| | | MET-min/week | Physical activity-related energy expenditure (MET-min/week) which is calculated by summing the product of frequency, intensity, and duration |
| 2 | Questionnaire | Hypertension | Long-term high blood pressure |
| | | MMSE score | To measure cognitive impairment |
| 3 | Environmental characteristics | Insomnia score | To evaluate the severity of sleep disturbance during the past 2 weeks |
| | | QOL | Degree to which an individual is healthy, comfortable, and able to participate in or enjoy life events |
| 4 | Spatio-temporal parameters in tems of three different cases (slower, preferred, and faster) | walking speed | Measure appropriate for assessing and monitoring functional status and overall health in a wide range of populations |
| | | stride length | Distance measured parallel to the Line of Progression, between the Posterior Heel points of two consecutive footprints of the foot in question |
| | | stance phase | Period of time that the foot is on the ground |
| | | CV stride length | Coefficient of variation for stride length |
| | | CV stance phase | Coefficient of variation for stance phase |
| | | gait asymmetry | To evaluate how leg movements differ while walking (comparing swing times between the legs and symmetry of swing duration) |
| | | cadence | A total number of full cycles taken within a given period of time, often expressed in steps or cycles per minute |
| | | stride time | An amount of time from consecutive heel contacts of the same foot. |
| | | CV stride time | Coefficient of variation for stride time |
| 5 | Physical fitness variables | grip strength | Force applied by the hand to pull on or suspend from objects and is a specific part of hand strength |
| | | five times STS | To measure one aspect of transfer skill |
| | | bicepts curl | A number of weight training exercises |
| | | chair sit and reach | One of the linear flexibility tests which helps to measure the extensibility of the hamstrings and lower back |
| | | three meter TUG | To determine fall risk and measure the progress of balance |
| | | back scratch | The reciprocal exchange of favors, services, assistance, or praise. |
| | | single leg balance | To measure the length of time the subject can maintain their balance |
| | | six min walk | To assess aerobic capacity and endurance |
| | | standing time from a long sitting position | Sitting behind your desk all day is bad for your health and experts have long been advising people to stand at their workstations for about 15 minutes an hour. |

**Figure 3.** Walking-related features extracted by the non-invasive method.

## 3. Classification through Automated Machine Learning

Most AutoML tools follow a typical three-step pipeline described in Figure 4, which shows typical components of a machine learning problem pipeline [28]. The first step consists of preparing the data [29]. This step involves loading and cleaning the data for use in the system and applying any transformations, normalizations, or encodings. The next step involves selecting to select the model [30]. This step might also involve feature engineering, which uses domain knowledge to generate new features to support and improve the machine learning model. Then, the final step consists of an iterative process in which one builds, trains, optimizes, validates, and selects a given machine learning algorithm to use for a given problem. In general, these three components are optimized iteratively to obtain the best results.

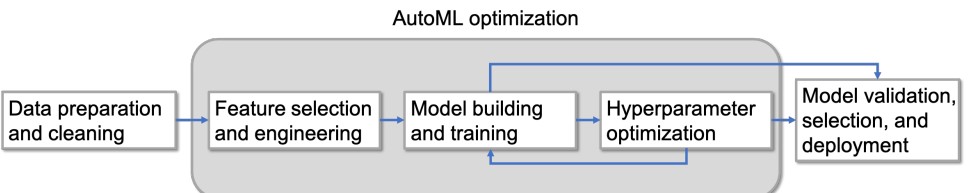

**Figure 4.** Important components and process of a machine learning problem pipeline.

The function of $f : \mathcal{X} \to \mathcal{Y}$ is good for the learning stage. The classification or regression uses $y$. An algorithm of $A$ can be set to $\{d_1, \cdots, d_n\}$ of training data points $d_i = (x_i, y_i) \in \mathcal{X} \times \mathcal{Y}$, which represents a parameter vector, and hyperparameters of $\lambda \in \Lambda$,

which represents changeing the method of the algorithm of $A_\lambda$. Here, hyperparameters indicate the length penalty, the number of neurons in a hidden layer, and the number of data in a decision tree. The loop can evaluate the performance of each hyperparameter configuration, which the cross-validation optimizes.

### 3.1. Data Preparation and Feature Engineering

Data pre-processing still requires considerable human intervention. This stage asks about the data type and schema detection. Thus, this work has not been primarily assisted among the AutoML. However, when one data type is identified, the tools support appropriate feature engineering.

### 3.2. Model Selection

After many different models use the driven features for training with different parameters, we can find the most proper model for the selection. When the algorithm of $\mathcal{A}$ and the limited amount of training data $\mathcal{D} = \{(x_1, y_1), \cdots, (x_n, y_n)\}$ are given, the model is to determine $\mathcal{A}^* \in \mathcal{A}$. The performance of each model is estimated by $\mathcal{D}$ into sets of $\mathcal{D}^{(i)}_{train}$ and $\mathcal{D}^{(i)}_{valid}$, $f_i$, with $\mathcal{A}^*$ to $\mathcal{D}^{(i)}_{train}$, and effect of the performance on $\mathcal{D}^{(i)}_{valid}$. These allow for the model selection problem as follows:

$$A^* \in \underset{A \in \mathcal{A}}{\arg\min} \frac{1}{k} \Sigma_{i=1}^{k} \mathcal{L}(A, \mathcal{D}^{(i)}_{train}, \mathcal{D}^{(i)}_{valid}), \tag{1}$$

where $\mathcal{L}(A, \mathcal{D}^{(i)}_{train}, \mathcal{D}^{(i)}_{valid})$ is the loss achieved by $A$ when trained on $\mathcal{D}^{(i)}_{train}$ and evaluated on $\mathcal{D}^{(i)}_{valid}$.

We use $k$-fold cross-validation, which splits the training data into $k$ equal-sized partitions $\mathcal{D}^{(1)}_{valid}, \cdots, \mathcal{D}^{(k)}_{valid}$, and sets $\mathcal{D}^{(i)}_{train} = \mathcal{D}/\mathcal{D}^{(i)valid}$ for $i = 1, \cdots, k$ [31].

### 3.3. Hyperparameter Optimization

The optimization of $\lambda \in \Lambda$ of $A$ resembles the model selection. It is possible to exploit the correlation structure between different settings of $\lambda_1, \lambda_2 \in \Lambda$. When $n$ hyperparameters $\lambda_1, \cdots, \lambda_n$ with domains $\Lambda_1, \cdots, \Lambda_n$ are given, the space of $\Lambda$ is a subset of the cross product: $\mathbf{\Lambda} \subset \Lambda_1 \times \cdots \times \Lambda_n$.

Hyperparameters of $\lambda_i$ can be replaced with another hyperparameter $\lambda_j$ when $\lambda_i$ is only active [32]. At that time, hyperparameter $\lambda_j$ takes values from a given set $\mathcal{V}_i(j) \not\subseteq \Lambda_j$; in this case, we call $\lambda_j$ a parent of $\lambda_i$. Conditional hyperparameters may be parents of other conditional hyperparameters [33]. When a structure of $\Lambda$ is given, the optimization can be solved as:

$$\lambda^* \in \underset{\lambda \in \Lambda}{\arg\min} \frac{1}{k} \Sigma_{i=1}^{k} \mathcal{L}(A_\lambda, \mathcal{D}^{(i)}_{train}, \mathcal{D}^{(i)}_{valid}). \tag{2}$$

## 4. Results of Dataset and Classification

### 4.1. Results of Constructed Dataset

Figure 5 shows an example of the constructed dataset for this study. As in the principal component analysis (PCA), the statistical method did not work well because the total number of feature vectors was 43. Questionnaires for seven feature vectors give a subjective judgment. However, that provides preliminary evidence that the interview with self-care and regimen adherence is a reliable and valid instrument and efficiently assesses self-care behaviors associated with glycemic control. The used features for this study are related to physical activities in daily life. The remaining 36 feature vectors are related to physical fitness and gait characteristics under imposed challenged speed conditions in senior citizens with diabetes during walking. The dimension of the matrix is (human subjects × feature vectors = 200 × 43).

| No | Feature | sb1 | sb2 | sb3 |
|----|---------|-----|-----|-----|
| 1 | age | 71 | 67 | 71 |
| 2 | gender | 1 | 0 | 1 |
| 3 | education_level | 1 | 3 | 1 |
| 4 | BMI | 27.9 | 27.9 | 28.5 |
| 5 | total_physical_activity_MET_min_week | 2895 | 6972 | 99 |
| 6 | hypertension | 0 | 1 | 1 |
| 7 | MMSE_score | 26 | 27 | 30 |
| 8 | slow_walking_speed_mps | 0.9 | 1.2 | 1.1 |
| 9 | slow_stride_length_m | 1.0 | 1.3 | 1.2 |
| 10 | slow_stance_phase_percent | 58.8 | 58.6 | 58.1 |
| 11 | slow_CV_stride_length_percent | 2.3 | 0.9 | 1.8 |
| 12 | slow_CV_stance_phase_percent | 3.3 | 2.3 | 2.3 |
| 13 | slow_gait_asymmetry_percent | 0.7 | 6.5 | 1.1 |
| 14 | slow_cadence_beats_min | 110.0 | 107.0 | 118.0 |
| 15 | slow_stride_time_s | 1.1 | 1.1 | 1.0 |
| 16 | slow_CV_stride_time_percent | 2.3 | 0.9 | 1.8 |
| 17 | preferred_walking_speed_mps | 1.0 | 1.5 | 1.3 |
| 18 | preferred_stride_length_m | 1.1 | 1.3 | 1.3 |
| 19 | preferred_stance_phase_percent | 58.4 | 57.5 | 56.8 |
| 20 | preferred_CV stride_length_percent | 2.9 | 2.1 | 1.0 |
| 21 | preferred_CV_stance phase_percent | 5.7 | 2.3 | 1.5 |
| 22 | preferred_gait_asymmetry_percent | 0.6 | 15.9 | 2.1 |

| No | Feature | sb1 | sb2 | sb3 |
|----|---------|-----|-----|-----|
| 23 | preferred_cadence_beats_min | 118.0 | 134.0 | 127.0 |
| 24 | preferred_stride_time_s | 1.0 | 0.9 | 1.0 |
| 25 | preferred_CV_stride_time_percent | 2.9 | 2.1 | 1.0 |
| 26 | fast_walking_speed_mps | 1.3 | 1.7 | 1.6 |
| 27 | fast_stride_length_m | 1.2 | 1.4 | 1.4 |
| 28 | fast_stance_phase_percent | 57.1 | 55.7 | 55.4 |
| 29 | fast_CV_stride_length_percent | 2.1 | 1.8 | 2.3 |
| 30 | fast_CV_stance_phase_percent | 3.0 | 1.9 | 2.8 |
| 31 | fast_gait_asymmetry_percent | 0.3 | 14.0 | 1.3 |
| 32 | fast_cadence_beats_min | 128.0 | 149.0 | 138.0 |
| 33 | fast_stride_time_s | 0.9 | 0.8 | 0.9 |
| 34 | fast_CV_stride_time_percent | 2.1 | 1.8 | 2.3 |
| 35 | grip_strength_kg | 24.0 | 32.9 | 25.8 |
| 36 | five_times_sit_to_stand_s | 13.0 | 14.7 | 9.2 |
| 37 | biceps_curl_reps | 22.0 | 19.0 | 22.0 |
| 38 | chair_sit_and_reach_cm | 19.4 | 10.5 | 20.6 |
| 39 | three_meter_TUG_s | 9.7 | 9.5 | 8.5 |
| 40 | back_scratch_cm | -2.0 | -11.3 | -8.0 |
| 41 | single_leg_balance_s | 2.7 | 5.1 | 95.0 |
| 42 | six_min_walk_s | 360.0 | 515.0 | 395.0 |
| 43 | standing_time_from_a_long_sitting_position_s | 3.63 | 4.15 | 9.70 |

**Figure 5.** An example of constructed dataset for this study.

Figure 6 shows the results of data distribution under the condition of three different walking speeds. Blue-colored data (mean $\pm$ standard deviation (SD) = 0.898 $\pm$ 0.155) represent the condition of slow speed, which becomes, on average, 22% slower walking speed than the preferred speed, and orange-colored data (1.157 $\pm$ 0.219) represent the preferred condition. Finally, gray-colored data (1.441 $\pm$ 0.275) represent the fast condition, which becomes the averagely 25% faster speed than the preferred, respectively.

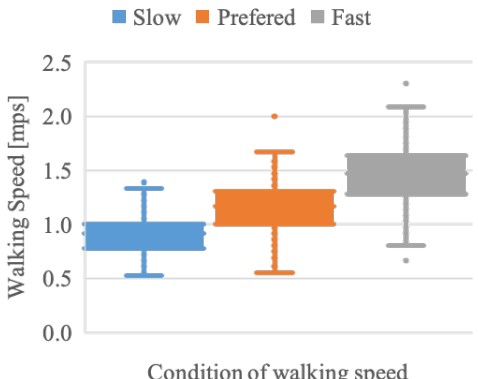

**Figure 6.** Results of data distribution under the condition of three different walking speeds: slow walking, preferred condition, and fast walking.

As a result, it was confirmed that there was no problem analyzing gait and physical fitness characteristics with the constructed dataset, although the experimental condition for walking speed was dependent on personal subjective criteria.

### 4.2. Results of Classification

Figure 7 shows the results of training through AutoML. The current AutoML trains and cross-validates many kinds of algorithms: XGBoost GBM (Gradient Boosting Machine), GLMs, default Random Forest (DRF), H2O GBMs, Deep Neural Net, Randomized Forest (XRT), and XGBoost GBMs. Thus, there is no need to consider which algorithms are proper for the prepared dataset because AutoML automatically looks into the results of all algorithms.

| model_id | auc | logloss | mean_per_class_error | rmse | mse |
|---|---|---|---|---|---|
| DRF_1_AutoML_20210810_015052 | 0.803 | 0.593 | 0.484 | 0.451 | 0.204 |
| GBM_3_AutoML_20210810_015052 | 0.776 | 0.632 | 0.440 | 0.462 | 0.214 |
| GBM_4_AutoML_20210810_01505 | 0.766 | 0.639 | 0.444 | 0.466 | 0.217 |
| StackedEnsemble_AllModels_AutoML_20210810_015052 | 0.758 | 0.607 | 0.500 | 0.457 | 0.209 |
| GBM_2_AutoML_20210810_015052 | 0.769 | 0.642 | 0.459 | 0.467 | 0.218 |
| StackedEnsemble_BestOfFamily_AutoML_20210810_015052 | 0.774 | 0.610 | 0.490 | 0.459 | 0.211 |
| GBM_1_AutoML_20210810_015052 | 0.755 | 0.754 | 0.500 | 0.496 | 0.246 |
| XGBoost_3_AutoML_20210810_015052 | 0.751 | 0.719 | 0.470 | 0.492 | 0.240 |
| GLM_1_AutoML_20210810_01505 | 0.749 | 0.599 | 0.464 | 0.453 | 0.205 |
| DeepLearning_1_AutoML_20210810_015052 | 0.699 | 0.759 | 0.500 | 0.502 | 0.252 |
| XGBoost_1_AutoML_20210810_015052 | 0.738 | 0.680 | 0.500 | 0.485 | 0.235 |
| XGBoost_2_AutoML_20210810_015052 | 0.681 | 0.619 | 0.500 | 0.462 | 0.213 |

**Figure 7.** Results of training through automated machine learning (AutoML). "auc" is related to the accuracy results of classification produced by the trained model, and "logloss" is that the cross-entropy between the model and the target values. "rmse" is the root-mean-square error metric, and "mse" is the mean square error. "mean_per_class_error" is one kind of available options for classification.

The training results found that it was possible to classify the difference in gait and physical fitness characteristics between senior citizens with or without diabetes with 80% accuracy through the non-invasive method.

Figure 8 shows the results of the confusion matrix for the prediction through DRF, which is the highest accuracy in used algorithms. Among the total data, 165 were used for training (80%), and 35 were used for prediction (20%). It was confirmed that the error rate was so low. That means that the applied algorithm was suitable for classifying the difference between senior citizens with and without diabetes.

**DRF_1_AutoML_20210810_015052**
Confusion Matrix (Act/Pred) for max f1 @ threshold = 0.3846153846153846:

| | | 0 | 1 | Error | Rate |
|---|---|---|---|---|---|
| **0** | 0 | 9.0 | 1.0 | 0.120 | (1.0/10.0) |
| **1** | 1 | 4.0 | 21.0 | 0.017 | (4.0/25.0) |
| **2** | Total | 13.0 | 22.0 | 0.143 | (5.0/35.0) |

**Figure 8.** Results of confusion matrix for the prediction through the default random forest (DRF), which is the highest accuracy in used algorithms.

Figure 9 shows the results of the variable importance plot, which show the relative importance of the essential variables in the model. Variable importance is currently available for all H2O models; so, if you happen to use h2o.explain() [34] on an AutoML object with a Stacked Ensemble at the top of the leaderboard, it instead shows the variable importance for the top "base model", which is DRF for this study, as in Figure 7.

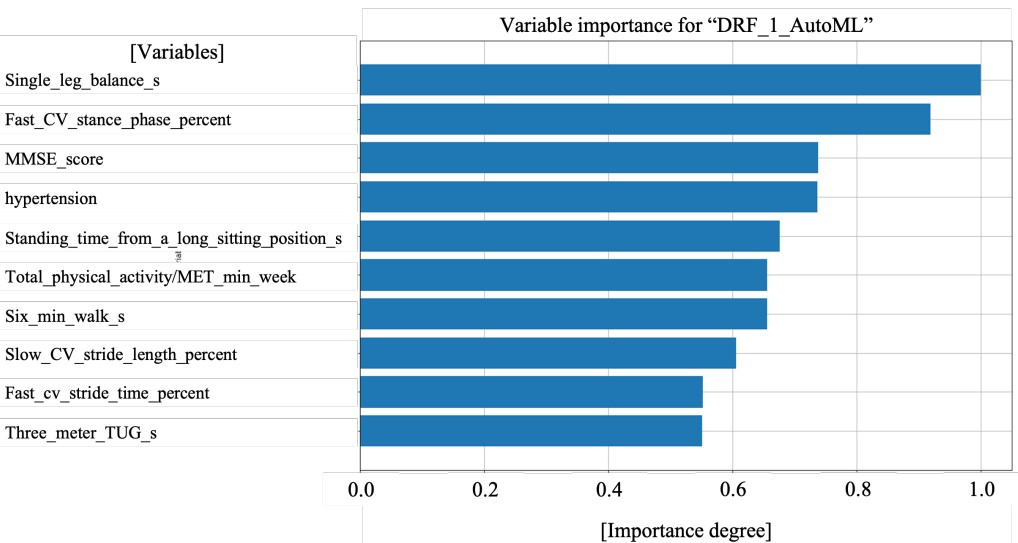

**Figure 9.** Results of variable importance plot, which show the relative importance of the most important variables in the model.

As a result, it was found that the variable of single_leg_balance_s was the most dominant. That meant that the balance stability might decrease more with senior citizens with diabetes than those without diabetes. Then, we found that senior citizens exhibited more insufficient gait stability at slower and faster strides. Thus, the different walking speeds must help evaluate gait characteristics to distinguish senior citizens with diabetes and controls. Additional important variables in the model were fast_CV_stance_phase_percent, standing_time_from_a_long_sitting_position_s, six_min_walk_s, slow_CV_stride_length_percent, fast_CV_stride_time_percent, three_meter_TUG_s, and non-physical performance variables, such as MMSE_score, hypertension, and total_physical_activity/MET_min_week. Variability (CV) domain of gait exhibited to be an important factor in senior citizens with diabetes.

## 5. Development of Machine-Learning-Based Personal Training System

Our study analyzing the gait and physical fitness of senior citizens based on machine learning found that the symmetry between the left and right feet differed in a fast walking speed because elderly patients with diabetes had a worse balance than healthy elderly adults. Therefore, although elderly adults have to exercise alone due to social distance under COVID-19 (pandemic situation), they need to train and evaluate their balance ability.

The authors have become interested in developing a home training system with an algorithm that allows users to evaluate their exercise poses alone at home using low-cost available devices. This study confirms the feasibility of an algorithm to evaluate the dynamic exercise pose using a low-cost single RGB camera, instead of an IMU sensor, as the gold standard, because the analysis of IMU sensors requires technical knowledge. Furthermore, the skeleton includes the information of movement of lower limbs, as well as others. The cheap USB-connected-typed RGB camera is the device that anyone can quickly obtain. Recent advances in technology, such as OpenPose, which can extract each joint from the human body based on human pose estimation, have been dazzling. However, to evaluate the exercise pose, the only key point representing each joint is not sufficient. Therefore, this study develops an algorithm to evaluate the exercise pose by comparing two skeletons for experts and users.

### 5.1. Experimental Condition

The experimental system in this study is composed of the RGB camera for acquiring data and a computer for processing and analyzing data. There is no problem using a high-resolution camera by connecting to the computer with the USB connector when the camera

is not built into the computer. The resolution of the camera in this study is 640 × 480. Some downloadable videos on the website construct a dataset of experts. Although there are many exercise activities for home training, we adapt yoga and squat, which are frequently used for home training.

Human pose estimation is performed by MediaPipe Pose, developed by Google [35]. MediaPipe Pose is an ML solution for high-fidelity body pose tracking with 33 3D joints on the whole body from RGB video frames utilizing BlazePose research that also powers the ML Kit Pose Detection API. Current advanced approaches rely essentially on powerful desktop environments for inference, whereas this method achieves real-time performance on most modern mobile phones, desktops/laptops, in Python, and even on the web [35].

Figure 10 represents a flow chart to explain how to process the image from the RGB camera and then to verify the similarity in exercise activities between experts and users in this study. It is necessary to perform camera calibration at first. Camera calibration is the process of estimating intrinsic or extrinsic parameters. Extrinsic parameters are mainly related to the position and orientation in the world, and intrinsic parameters deal with the camera's internal characteristics, such as its focal length, skew, distortion, and image center. Thus, we can say that intrinsic parameter is an essential first step for 3D computer vision, as it allows us to estimate the scene's structure in Euclidean space and removes lens distortion, which degrades accuracy. Then, it is necessary to reduce noise signals in time-series data. The location for each key point through human pose estimation technology based on a convolutional neural network (CNN) is time-series data which means the x-axis for the sampling time and y-axis for location value. In time series forecasting, the presence of dirty and messy data can hurt the final predictions. Therefore, temporal dependency plays a crucial role when dealing with temporal sequences. After two phases are completed, now, we are ready to calculate each link length (mean ± standard deviation(SD)) when the user shows the stationary pose. Moreover, finally, it is necessary to evaluate the similarity in exercise activities between experts and users after selecting the kind of exercise activities. Human coaches are very good at visually detecting such patterns, although trainees show performance with different speeds. Nevertheless, programming machines to do the same is a complex problem. Successful recognition strategies are based on the ability to approximately match amplitude for each key point, despite wide variations in timing.

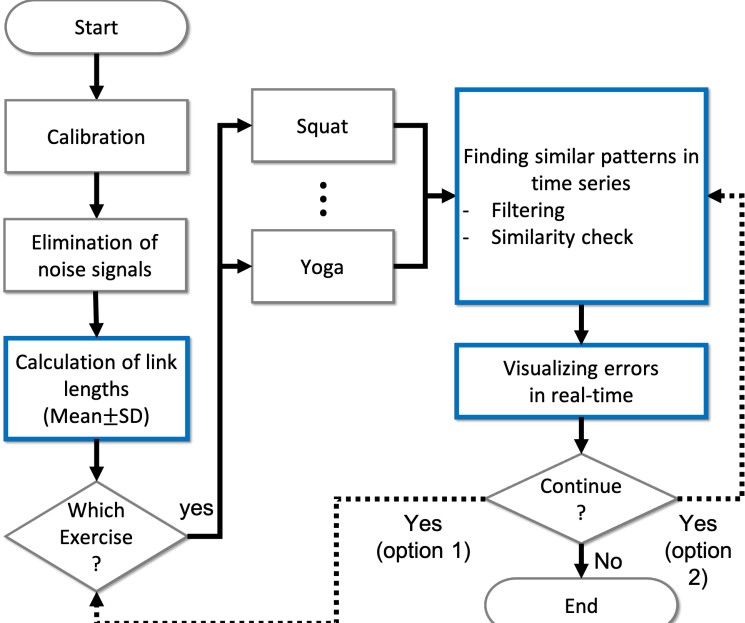

**Figure 10.** A flowchart to explain how to process the image acquired from the RGB camera and then to verify the similarity in exercise activities between experts and users.

*5.2. Dynamic Time Warping Algorithm for Similarity Evaluations*

In particular, extracting some features in continuously measured signals seems to include many essential aspects of pattern detection in time series. Feature extraction is usually based on matching templates against a waveform of the continuous signal, converted into a discrete-time series. Thus, successful recognition strategies are based on matching signals, despite wide variation in timing and amplitude, approximately. Because the speed of exercise activity for experts is different from that for the user, it is necessary to evaluate the similarity between two different time-span data. Any two-time series can be compared using Euclidean distance or similar distances on a one-to-one basis on the time axis. The amplitude of first time-series data at time $T$ is compared with the amplitude of second time-series data at time $T$. The comparison at the same time axis leads to an inferior comparison and similarity score even if the shape of the two-time series are very similar but out of phase in time. Dynamic time warping (DTW) compares amplitude of first signal at time $T$ with amplitude of second signal at time $T + 1$ and $T - 1$ or $T + 2$ and $T - 2$. This makes sure it does not give low similarity score for signals with similar shape and different phase [36,37].

The DTW technique is based on an approach of dynamic programming, while aligning the time series to find minimized distance measurement. Because some regulations of the time axis can fit the horizontal axis, the proper template looks useful. Investigating a time series, $S$, is related to finding the pattern in a template, $T$,

$$S = s_1, s_2, \cdots, s_i, \cdots, s_n \tag{3}$$

$$T = t_1, t_2, \cdots, t_j, \cdots, t_m. \tag{4}$$

The sequences $S$ and $T$ have a $n - by - m$ matrix. Each element of $(i, j)$ at the matrix represents the similarity between the two elements of $s_i$ and $t_j$. $W$ aligns the elements of $S$ and $T$, and the warping path has the minimum value,

$$W = w_1, w_2, \cdots, w_k, \tag{5}$$

where $W$ represents a sequential point, and $w_k$ represents $(i, j)_k$.

Solving a dynamic programming problem is to have a measured distance between two elements. Although we have many candidates, it looks proper for the absolute value or square of the similarity as the distance function of $\delta$.

$$\delta(i, j) = |s_i - t_j| \tag{6}$$

$$\delta(i, j) = (s_i - t_j)^2. \tag{7}$$

The function of $\delta$ means a measured distance between two-time series data. Since the cumulative measurement for each path indicates the potential warping paths, the DTW problem can define as minimization of warping paths.

$$DTW(S, T) = \min_W \left[ \Sigma_{k=1}^p \delta(w_k) \right]. \tag{8}$$

Dynamic programming explains legal state transitions with stage, state, and decision. Although the decision is difficult to recognize, these variables show possible paths between the two elements in the matrix. Some limitations are as follows. However, these are good for deciding permissible paths for efficiency.

1.　Monotonicity:
　　The points must be monotonically ordered with respect to time, $i_{k-1} \leq i_k$ and $j_{k-1} \leq j_k$.
2.　Continuity:
　　The steps in the grid are confined to neighboring points, $i_k - i_{k-1} \leq 1$ and $j_k - j_{k-1} \leq 1$.

3. Warping window:
   Allowable points can be constrained to fall within a given warping window, $|i_k - j_k| \le w$, where $w$ is a positive integer window width.
4. Slope constraint:
   Allowable warping paths can be constrained by restricting the slope, avoiding extensive movements in a single direction.
5. Boundary conditions:
   The starting point selects one of the subsequent paths, and the endpoint adds some offset to constrained points, such as $i_1 = 1$, $j_1 = 1$ and $i_k = n$, $j_k = m$.

The dynamic programming algorithm is based on the following recurrence relation, which defines the cumulative distance, $\gamma(i, j)$, for each point,

$$\gamma(i, j) = \delta(i, j) + \min[\gamma(i-1, j), \gamma(i-1, j-1), \gamma(i, j-1)]. \tag{9}$$

Filling the lowest cumulative distances in the matrix helps us find the optimal warping path.

## 6. Results of Machine-Learning-Based Personal Training System

### 6.1. Results of Graphical User Interface

Figure 11 shows a description of the graphical user interface (GUI) for the developed fitness software. The current pose of the user is to initialize each link length according to the key point detection. The initialization results make it possible to calculate the value of (mean ± standard deviation (SD)) for each link length on the approximate 3D space. At the bottom of the user picture, "Reps" indicates the counter for the number of exercise activities, "Feedback" indicates the advice for the excellent exercise pose, and "Timer" indicates the history for the exercise time, respectively. On the right side of the figure, the bar plot represents the achievement per one exercise activity.

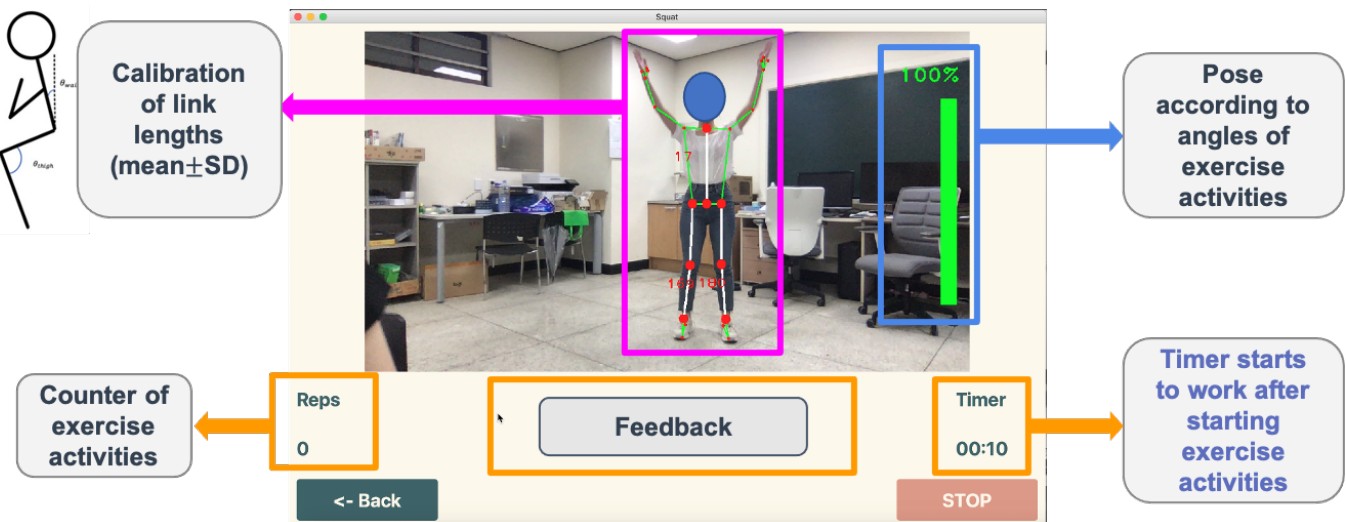

**Figure 11.** A description of graphical user interface (GUI) for the developed fitness software.

### 6.2. Results of Similarity Evaluation through Dynamic Time Warping Algorithm

Figure 12 shows results of similarity in time series data through DTW algorithm. The horizontal axis, as shown in the left-side plot in Figure 12b,c, indicates the number of frames which is similar to the sampled time, and the vertical axis indicates calculated angles of $\theta_{trunk}$ and $\theta_{leg}$, as shown in Figure 11, on the left. The orange curved plot of "pro" results from training experts, and the blue curve of "nov" is the users. The user who did not have much experience with this fitness system hesitated to begin during 40 frames of 1.3 s because she had no idea when to start. Many lines connecting one point on the

curve for experts to one or several points on the curve for the user are results of DTW comparison. DTW compares amplitude of first signal at time $T$ with amplitude of second signal at time $T + 1$ and $T - 1$ or $T + 2$ and $T - 2$. The comparison with the changing time axis makes sure it does not give a low similarity score for signals with similar shapes and different phases.

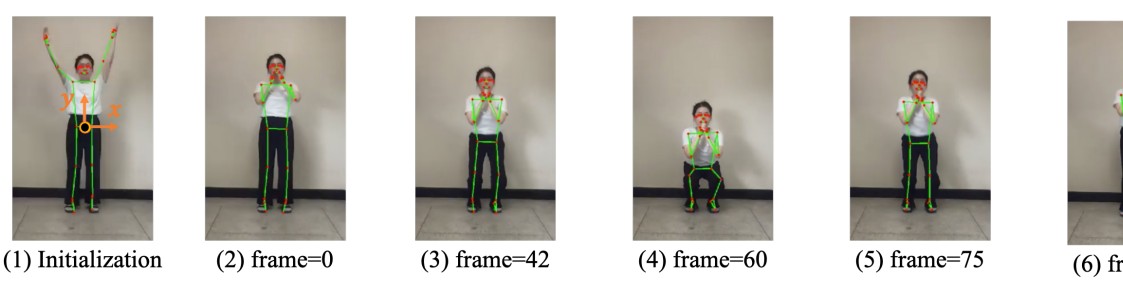

(1) Initialization　　(2) frame=0　　(3) frame=42　　(4) frame=60　　(5) frame=75　　(6) frame=85

(a) Results of image data including extracted key points through human pose estimation.

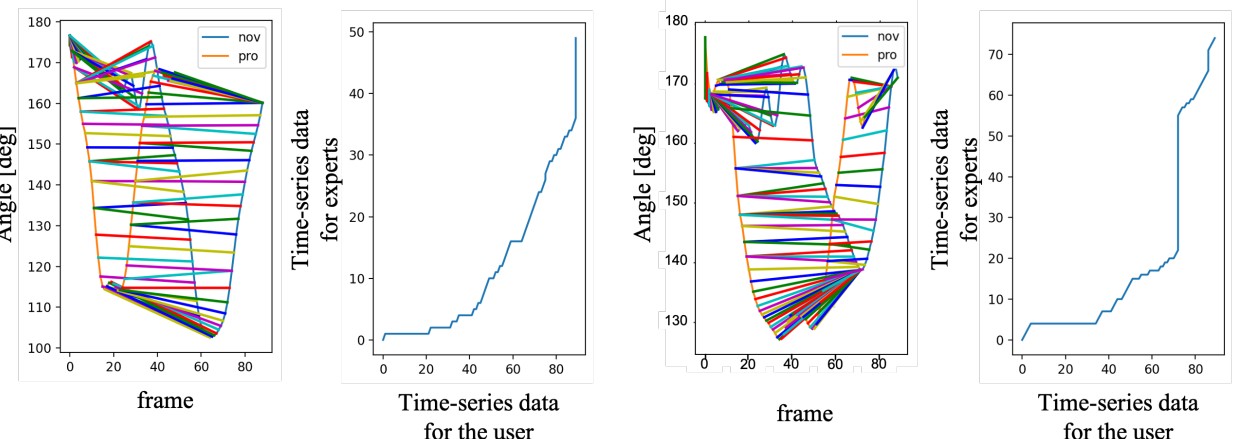

(b) Results of time series similarity of the angle of trunk using dynamic time warping.

(c) Results of time series similarity for the angle of lower limb using dynamic time warping.

**Figure 12.** Results of similarity in time series data through dynamic time warping (DTW). (**a**) represents image data including extracted key points through human pose estimation, and (**b**,**c**) represent similarity in time series data for angles of the trunk and lower limbs, respectively.

The right-side plot in Figure 12b,c shows the results of the cost matrix and warping path: the horizontal axis represents data for the user, and the vertical axis represents those for averaged experts. The closer the plot is to the diagonal, the higher the exercise activity for the user is similar. The closer the plot is to the horizontal axis, the lower the activity for the user is the similarity. The line shows the zero DTW distance. Although the user was late to start, it was found that there was no problem evaluating similarities in time series data of exercise activities through the DTW algorithm.

Figure 13 shows results of visualized error points in the real-time through DTW algorithm. As a result, it was proven that the DTW algorithm helped evaluate exercise activities. Furthermore, this system could monitor us and provide real-time feedback if we extended our knees too far or our legs were placed too close.

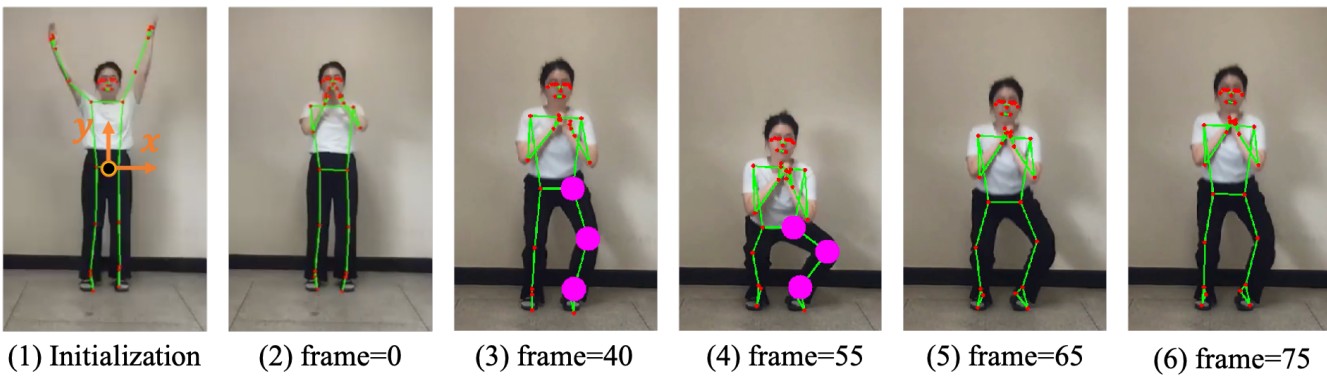

(1) Initialization  (2) frame=0  (3) frame=40  (4) frame=55  (5) frame=65  (6) frame=75

(a) Results of image data of lower limbs including error points through dynamic time warping (DTW).

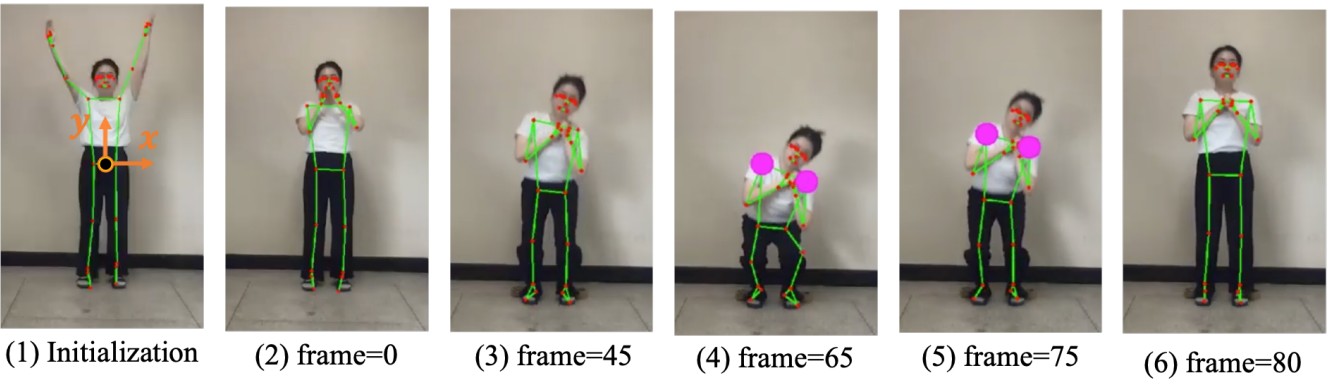

(1) Initialization  (2) frame=0  (3) frame=45  (4) frame=65  (5) frame=75  (6) frame=80

(b) Results of image data of shoulders including error points through dynamic time warping (DTW).

**Figure 13.** Results of visualized error points in the real-time through dynamic time warping (DTW): (**a**) Errors in the left lower limb; (**b**) errors in both shoulders. This function to visualize errors in real-time helps correct the poor exercise pose of the user by herself.

## 7. Discussions

Although most people want home training systems to replace human coaches, methods are still not perfect. Therefore, it is necessary to know what kinds of failure cases are still existed. Among many different exercise activities, squats have become an excellent example of applying for human pose estimation technology and have been proper for describing widespread errors resulting in severe healthcare problems. While some athletes perform power-lifting, the most common exercise, most athletes request a personal coach because the heavyweight exercise tool produces poor posture. Then, it is time to clarify whether home training systems based on human pose estimation can substitute human training coaches or not.

(1) Body specifics according to the gender:
When image data of humans train human-pose-estimation-based models, it is necessary to consider the difference in physiology between males and females. For example, if the dataset for the train includes many men's images, the accuracy for the prediction depends on only male users. Meanwhile, if women's users use the trained model, wrong results wait for us, although the exercise posture is good. Thus, the model should consider the difference between the two genders when the home training system is developed based on human pose estimation.

(2) Physiology specifics:
The model based on human pose estimation can recognize the user's body through the dataset of image data. However, it is not easy to guarantee whether images for training are similar body structures or not. Thus, the prediction results are always

low even if the exercise posture is correct when the training does not use the dataset with the general proportion for the body parts.

(3) Decision of the exercise start:

There is no problem that the user follows human coaches to start and finish the exercise. However, it is difficult to tell the home training system when the activity begins and ends. Thus, the dataset should be time-series data because it doubts to decide the exercise duration with some images.

(4) Decision of frontal view:

When we need to compare exercise postures with two different videos, there is no guarantee which the taken conditions of the camera, such as the angle, height, and lighting, are similar. Thus, finding the frontal view from the recorded video is always impossible, and the results may be insufficient. The dataset still does not contain enough information for alignment.

(5) Problem for quick movements of the body part:

The frame rate of the web camera is 30 or 60 Hz, in general. That means that the model based on human pose estimation does not allow fast movements for exercises to detect exact key points. Although deep learning improves pose estimation technology, blurred image data are not helpful for training.

(6) Decision of horizontal position:

It is not easy to find the flat plane from the image. Thus, it is difficult to find the horizontal and vertical axes from the only image when someone performs exercises. That is the reason why we should need to calibrate images before training.

(7) Decision of occluded joints:

The occlusion problem is a problematic issue when finding key points from the image through human pose estimation. According to the taken condition, some bodies and objects hide target joints, in general. At that time, it is necessary to decide how to estimate hidden or lost joints. However, there is still not a clear to solve the occlusion problem.

## 8. Conclusions

This study addressed two issues: (1) use sensor data and AI/ML to classify senior citizens with diabetes based on their gait and physical fitness characteristics and (2) develop a personal training program using AI/DL based on 3D skeleton detection. Thus, using IMU sensor data and ML, we could classify the elderly with diabetes based on their gait and physical fitness characteristics and learned how to develop a personal training program using AI/DL, e.g., 3D skeleton detection.

In fact, this study aimed to prove that abnormalities for senior citizens with diabetes were classified under imposed challenge walking speed conditions; slow (=22% slower than preferred speed), preferred, and fast (=25% faster than preferred speed) walking speeds through AutoML. The dataset for training was constructed with the support of senior citizens in the community by using the IMU. The applied AutoML for classification is an emerging research field within computer science that can help non-experts use machine learning off the shelf. Furthermore, the developed ML-based personal home training system using the single RGB camera showed the high possibility of correcting the exercise posture and speed in real-time. Therefore, the results of this study may be helpful for the self-preservation of senior citizens with diabetes by themselves with a single RGB camera.

**Author Contributions:** Conceptualization, B.N. and C.Y.; methodology, B.N., M.L., H.P. and H.J.; software, Y.W., S.K., S.A. and H.T.P.N.; validation, Y.W., S.K. and S.A.; formal analysis, Y.W. and H.J.; investigation, H.J.; resources, B.N., M.L., H.P. and C.Y.; writing—original draft preparation, H.J.; writing—review and editing, H.T.P.N., H.J. and B.N.; visualization, S.K. and H.J.; supervision, C.S. and C.Y. All authors have read and agreed to the published version of the manuscript.

**Funding:** This research was funded by the Basic Science Research Program through the National Research Foundation (NRF) of Korea grant funded by the Ministry of Education (No. 2021R1I1A305521011), and the Design Innovation Program (AHA Platform and Personalized Services for the Elderly Using Universal UX Design, (No. 20012692) funded By the Ministry of Trade, industry & Energy (MOTIE, Republic of Korea).

**Institutional Review Board Statement:** This study was conducted according to the guidelines of the Declaration of Helsinki, and approved by the Institutional Review Board of Dong-A University (IRB: 2-104709-AB-N-01-201808-HR-023-02).

**Informed Consent Statement:** Informed consent was obtained from all subjects involved in the study.

**Data Availability Statement:** The dataset is available for your non-commerical research. Please contact any author. You can download our dataset after you submit the document for your agreement.

**Conflicts of Interest:** The authors declare no conflict of interest.

## Abbreviations

The following abbreviations are used in this manuscript:

AutoML     Automated Machine Learning
DTW         Dynamic Time Warping

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
