# Peer review of "Classification of Diabetic Walking for Senior Citizens and Personal Home Training System Using Single RGB Camera through Machine Learning"

_applsci, doi:10.3390/app11199029_

Round 1

Reviewer 1 Report

This manuscript is well written and well presented. The motivation and methodology of using sensor data with machine learning to classify elderly with diabetes based on their gait and physical fitness characteristics were clearly stated and justified. Literature review is well done and supports the research goal. Based on the evaluation results, a personal training program is developed. The research described in this manuscript is complete and well planned. I recommend publication of this manuscript.

This research addresses two issues: 1) use sensor data and AI/ML to classify elderly with diabetes based on their gait and physical fitness characteristics, and 2) develop a personal training program using AI/DL, e.g., 3D skeleton detection.

I am aware of lots of plenty research work employing inertial sensor data and ML to classify healthy vs patients (e.g., Parkinson) based on their gait and physical fitness characteristics. But I am not aware of research work on using IMU and ML to classify elderly with diabetes based on their gait and physical fitness characteristics. And the authors justify this research direction in the introduction section with sufficient literature review.

We can classify elderly with diabetes based on their gait and physical fitness characteristics using IMU sensor data and ML, and 2) We learn how to develop a personal training program using AI/DL, e.g., 3D skeleton detection.

I think this paper is well written and ready to publish. The research group has done a good job. But if a minor modification is required, I would suggest the authors explain how to use the gait and physical fitness characteristics measured by IMU (Section 4) to develop a personal training program (Section 5). The authors do not explain clearly how the ML based assessment described in Section 4 is used as input to build the personal training program in Section 5.

The literature review is good and justifies the research motivation.

Author Response

Dear Reviewer #1

Thank you for your contribution. Your comments are so helpful for our study. It may not be enough, but we did our best to respond.

  • Comment #1

This manuscript is well written and well presented. The motivation and methodology of using sensor data with machine learning to classify elderly with diabetes based on their gait and physical fitness characteristics were clearly stated and justified. Literature review is well done and supports the research goal. Based on the evaluation results, a personal training program is developed. The research described in this manuscript is complete and well planned. I recommend publication of this manuscript.

  • Response #1

Thank you for your good evaluation.

  • Comment #2

This research addresses two issues: 1) use sensor data and AI/ML to classify elderly with diabetes based on their gait and physical fitness characteristics, and 2) develop a personal training program using AI/DL, e.g., 3D skeleton detection.

  • Response #2

Thank you for your good understanding.

  • Comment #3

I am aware of lots of plenty research work employing inertial sensor data and ML to classify healthy vs patients (e.g., Parkinson) based on their gait and physical fitness characteristics. But I am not aware of research work on using IMU and ML to classify elderly with diabetes based on their gait and physical fitness characteristics. And the authors justify this research direction in the introduction section with sufficient literature review.

  • Response #3

Thank you for your good reputation.

  • Comment #4

We can classify elderly with diabetes based on their gait and physical fitness characteristics using IMU sensor data and ML, and 2) We learn how to develop a personal training program using AI/DL, e.g., 3D skeleton detection.

I think this paper is well written and ready to publish. The research group has done a good job. But if a minor modification is required, I would suggest the authors explain how to use the gait and physical fitness characteristics measured by IMU (Section 4) to develop a personal training program (Section 5). The authors do not explain clearly how the ML based assessment described in Section 4 is used as input to build the personal training program in Section 5.

The literature review is good and justifies the research motivation. 

  • Response #4

Thank you for your support. This paper tried to find the classification and train the good exercise pose at home. The IMU sensor was used for the gold standard. Then the RGB camera was utilized for the practical use. Because IMU data requires technical knowledge, but images do not need. I added the following sentences at the first paragraph in 5.1 section.

Our study analyzing the gait and physical fitness of senior citizens based on machine learning found that the symmetry between the left and right feet differed in a fast walking speed because elderly patients with diabetes had a worse balance than healthy elderly adults. Therefore, although elderly adults have to exercise alone due to social distance under COVID-19 (pandemic situation), they need to train and evaluate their balance ability.

The authors have become interested in developing a home training system with an algorithm that allows users to evaluate their exercise poses alone at home using low-cost available devices. This study confirms the feasibility of an algorithm to evaluate the dynamic exercise pose using a low-cost single RGB camera instead of IMU sensor as the gold standard because the analysis of IMU sensors requires technical knowledge. Furthermore, the skeleton includes the information of movement of lower limbs as well as others. The cheap USB-connected-typed RGB camera is the device that anyone can quickly obtain. Recent advances in technology such as OpenPose, which can extract each joint from the human body based on human pose estimation, have been dazzling. However, to evaluate the exercise pose, the only key point representing each joint is not sufficient. Therefore, this study develops an algorithm to evaluate the exercise pose by comparing two skeletons for experts and users.

Thank you for your contribution in this work.

My Best Regards,

Hieyong Jeong, PhD (Engineering)

Chonnam National University

Homepage: https://sites.google.com/view/human-media-lab/home

Reviewer 2 Report

  • The title is too long, and a shorter version should be considered.
  • The authors should consider the wider spectrum of the literature and refer some technologies used for gait analysis and eHealth applications, both electronic and photonic based solutions recently reported. Some good examples that should be referenced are:
  1. "Non-invasive Wearable OpticalSensors for full Gait Analysis in e-Health Architectures". EEE Wireless Communications, Vol. 28, No. 3, pp. 28 – 35 (June 2021). doi: 10.1109/MWC.001.2000405
  2. “Insole optical fiber sensor architecture for remote gait analysis – an eHealth Solution”, IEEE Internet of Things Journal,  July, 2017. DOI: 1109/JIOT.2017.2723263
  • The text should proof read. Same typos and formatting mistakes should be corrected along the text: example: body in our body.(line24); tables appear cut (table 1).
  • Table 2 is presented without any reference to it, or analysis in the section that it is presented (2.2). That should be corrected, and table 2 should be moved to section 2.3
  • It should be added a paragraph, better explaining the contribution and the novelty of the work presented.
  • How many sensors, are were needed for the algorithm training? Only the IMU sensors? How does it translate in the final fitness software?
  • What equipment will the final user have to acquire, to be able to use the proposed software?
  • Conclusion is very vague, and should be further extended.

Author Response

Dear Reviewer #2

Thank you for your contribution. Your comments are so helpful for our study. It may not be enough, but we did our best to respond.

  • Comment #1

The title is too long, and a shorter version should be considered.

  • Respond #1

The main title is “Consideration of the machine-learning-based personal home training system for senior citizens with diabetes”. But in order to express more detail, I add the sub-title “From dataset construction to classification of gait and physical fitness characteristics and development of personal home training system using single RGB camera”.

As you mentioned, the total title looks long, but the issue of this paper is 1) use sensor data with machine learning and 2) development of a personal training system. 1) and 2) should be connected.

Thus, I would like to remain the full title if I can get your understanding.

  • Comment #2

The authors should consider the wider spectrum of the literature and refer some technologies used for gait analysis and eHealth applications, both electronic and photonic based solutions recently reported. Some good examples that should be referenced are:

"Non-invasive Wearable OpticalSensors for full Gait Analysis in e-Health Architectures". IEEE Wireless Communications, Vol. 28, No. 3, pp. 28 – 35 (June 2021). doi: 10.1109/MWC.001.2000405

“Insole optical fiber sensor architecture for remote gait analysis – an eHealth Solution”, IEEE Internet of Things Journal, July, 2017. DOI: 1109/JIOT.2017.2723263

  • Respond #2

Thank you for your supporting. I added two references of [14] and [15] in my manuscript.

  • Comment #3

The text should proof read. Same typos and formatting mistakes should be corrected along the text: example: body in our body. (line24); tables appear cut (table 1).

  • Respond #3

I am so sorry about our mistakes. I revised the line 24 to erase “in our body”.

But I could not understand your comment “tables appear cut (table 1)”. You meant that the table seemed that some parts was cut. If yes, this was the table style, not cut. Please let me know the exact meaning if my guess was wrong.

  • Comment #4

Table 2 is presented without any reference to it, or analysis in the section that it is presented (2.2). That should be corrected, and table 2 should be moved to section 2.3

  • Respond #4

Thank you for your kind comment. Table 2 was to represent the dataset construction, not to analyze the dataset. The results of dataset were shown in section 4.1. In section 2.3, the number of 43 measured variables in this study was utilized for finding the primary feature. Thus, I thought that it was necessary to describe the structure of dataset before analyzing it. The summary of dataset description allowed readers to understand what kinds of data were used to find primary featured vectors.

Table 2 was explained in section 2.3, not in section 2.2. So I thought that it was not necessary to move it.

  • Comment #5

It should be added a paragraph, better explaining the contribution and the novelty of the work presented.

  • Respond #5

Thank you for your opinion. But I almost prepared the contribution from the line 89 to the line 95. This study addressed two issues: 1) use sensor data and AL/ML to classify elderly with diabetes based on their gait and physical fitness characteristics, and 2) develop a personal training program using AL/DL, e.g., 3D skeleton detection.

  • Comment #6

How many sensors, are were needed for the algorithm training? Only the IMU sensors? How does it translate in the final fitness software?

  • Respond #6

Thank you for your comment. Two IMU sensors were used to attach the backside of shoe as shown in Figure 1. These data through IMU sensors were to decide the gold standard. Then, the next experiment confirmed the feasibility of an algorithm to evaluate the dynamic exercise pose using a low-cost single RGB camera instead of IMU sensor because the analysis of IMU sensors requires technical knowledge. Furthermore, the skeleton includes the information of movement of lower limbs as well as others.

I added the following sentences at the first paragraph in 5.1 section.

Our study analyzing the gait and physical fitness of senior citizens based on machine learning found that the symmetry between the left and right feet differed in a fast walking speed because elderly patients with diabetes had a worse balance than healthy elderly adults. Therefore, although elderly adults have to exercise alone due to social distance under COVID-19 (pandemic situation), they need to train and evaluate their balance ability.

The authors have become interested in developing a home training system with an algorithm that allows users to evaluate their exercise poses alone at home using low-cost available devices. This study confirms the feasibility of an algorithm to evaluate the dynamic exercise pose using a low-cost single RGB camera instead of IMU sensor as the gold standard because the analysis of IMU sensors requires technical knowledge. Furthermore, the skeleton includes the information of movement of lower limbs as well as others. The cheap USB-connected-typed RGB camera is the device that anyone can quickly obtain. Recent advances in technology such as OpenPose, which can extract each joint from the human body based on human pose estimation, have been dazzling. However, to evaluate the exercise pose, the only key point representing each joint is not sufficient. Therefore, this study develops an algorithm to evaluate the exercise pose by comparing two skeletons for experts and users.

  • Comment #7

What equipment will the final user have to acquire, to be able to use the proposed software?

  • Respond #7

Thank you for your detail comment. The final user can use the cheap single RGB camera for the personal home training system. The system enables the user to exercise the correct pose through the dynamic time warping (DTW) algorithm, compared with the exercise pose for experts.

  • Comment #8

Conclusion is very vague, and should be further extended.

  • Respond #8

Thank you for your comment. I added the issue and the short results. Then, I prepared the detailed conclusion of this study.

This study addressed two issues: 1) use sensor data and AI/ML to classify senior citizens with diabetes based on their gait and physical fitness characteristics, and 2) develop a personal training program using AI/DL based on 3D skeleton detection. Thus, using IMU sensor data and ML, we could classify the elderly with diabetes based on their gait and physical fitness characteristics and learned how to develop a personal training program using AI/DL, e.g., 3D skeleton detection.

In fact, this study aimed to prove that abnormalities for senior citizens with diabetes were classified under imposed challenge walking speed conditions; slow (=22\% slower than preferred speed), preferred, and fast (=25\% faster than preferred speed) walking speeds through AutoML. The dataset for training was constructed with the support of senior citizens in the community by using the IMU.

The applied AutoML for classification is an emerging research field within computer science that can help non-experts use machine learning off the shelf.

Furthermore, the developed ML-based personal home training system using the single RGB camera showed the high possibility of correcting the exercise posture and speed in real-time.

Therefore, the results of this study may be helpful for the self-preservation of senior citizens with diabetes by themselves with a single RGB camera.

Finally, I got the help for English Proof Reading with Grammarly (not free). Because there is a long vacation in Republic Korea from 18 to 22 September 2021, it is not easy to use the service of Editage. If you have serious trouble to read English, I will try the English Proof Reading from Editage.

Thank you for your contribution in this work.

My Best Regards,

Hieyong Jeong, PhD (Engineering)

Chonnam National University

Homepage: https://sites.google.com/view/human-media-lab/home

Round 2

Reviewer 2 Report

Although the authors provided the answer to all the concerns, the changes performed are not marked/highlighted in the manuscript. 

The authors should show the changes performed in the manuscript.

Author Response

Dear Reviewer #2

Thank you for your contribution. Your comments are so helpful for our study. It may not be enough, but we did our best to respond.

  • Comment #1

Although the authors provided the answer to all the concerns, the changes performed are not marked/highlighted in the manuscript. 

The authors should show the changes performed in the manuscript.

  • Response #1

I am so sorry about your inconvenience. I attached the revised version with highlighted lines. Please confirm it.

Thank you for your contribution in this work.

My Best Regards,

Hieyong Jeong, PhD (Engineering)

Chonnam National University

Homepage: https://sites.google.com/view/human-media-lab/home
